# SKD-NER: Continual Named Entity Recognition via Span-based Knowledge Distillation with Reinforcement Learning

**Yi Chen[1,2] , Liang He[1,2,3]*, Lei Wang[1,2] , Zhenxiang Han[1,2]**

[1]School of Computer Science and Technology, Xinjiang University, Urumqi 830017, China
[2]Xinjiang Key Laboratory of Signal Detection and Processing, Urumqi 830017, China
[3]Department of Electronic Engineering, and Beijing National Research Center for Information Science and Technology, Tsinghua University, Beijing 100084, China
Stephen2637@hotmail.com, heliang@tsinghua.edu.cn

## Abstract

Continual learning for named entity recognition (CL-NER) aims to enable models to continuously learn new entity types while retaining the ability to recognize previously learned ones. However, the current strategies fall short of effectively addressing the catastrophic forgetting of previously learned entity types. To tackle this issue, we propose the SKD-NER model, an efficient continual learning NER model based on the span-based approach, which innovatively incorporates reinforcement learning strategies to enhance the model's ability against catastrophic forgetting. Specifically, we leverage knowledge distillation (KD) to retain memory and employ reinforcement learning strategies during the KD process to optimize the soft labeling and distillation losses generated by the teacher model to effectively prevent catastrophic forgetting during continual learning. This approach effectively prevents or mitigates catastrophic forgetting during continuous learning, allowing the model to retain previously learned knowledge while acquiring new knowledge. Our experiments on two benchmark datasets demonstrate that our model significantly improves the performance of the CL-NER task, outperforming state-of-the-art methods.[1]

## 1 Introduction

The introduction of continual learning methods has enabled systems to continuously learn from new data and reduce their dependence on initial training data. Moreover, these methods facilitate model updates and fine-tuning, enhancing their scalability (Shin et al., 2017). By combining continual learning with NER tasks, systems can significantly improve their ability to perceive the constantly changing real world amidst the emergence of new tasks and data sources, these functionalities can be formulated as paradigms of continual learning (Jin

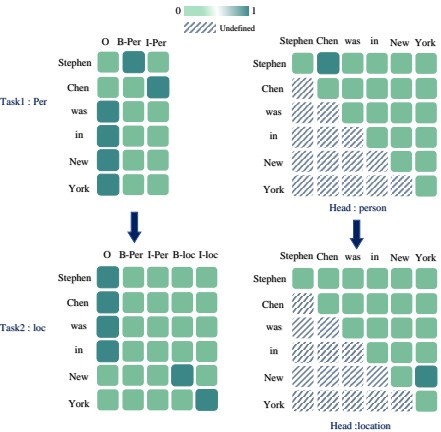

Figure 1: An illustration of the sequence labeling approach in SKD-NER. Assuming a model learns task1:Per and task2:Loc in OntoNotes5 sequentially, the left side represents the conventional sequence labeling method, while the right side represents the approach we adopted.

et al., 2022; Parisi et al., 2019). However, continual learning has always faced catastrophic forgetting, which has become a pervasive issue for continual learning NER tasks (McCloskey and Cohen, 1989; Robins, 1995; Kirkpatrick et al., 2017). Specifically, simply fine-tuning the NER system based on new data often results in a significant drop in performance on previously learned tasks, which poses a major challenge for achieving human-level intelligence in continual learning for NER (CL-NER). This is in contrast to the natural ability of humans to learn new entity categories without forgetting previously learned ones.

In the context of continual learning, the model training process is typically divided into $n$ CL steps, with each step being specific to the current task. In the case of CL-NER, only new entity types are recognized in each CL step. However, this approach can lead to a situation that is easily overlooked, whereby entity types that are not required to be recognized in the current step (e.g., ORG) may need to be learned in the future or have been learned in

---

*Corresponding author.
[1]Code is available at https://github.com/YChen2637/SKD

the past. In traditional sequence labeling methods,

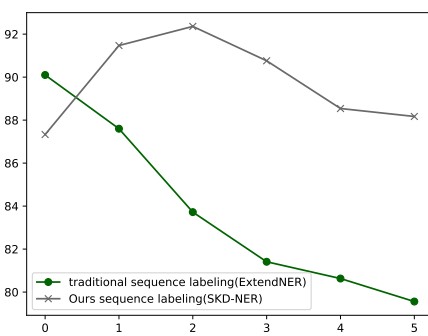

Figure 2: An illustration of the impact of sequence labeling methods on OntoNotes5.

an entity that is not a required type in the current step is assigned a global O tag to indicate that the model does not need to recognize this type in the current task(refer to Figure1). However, this can result in each entity's category needing frequent parameter updates in each different CL step, due to this incoherent optimization. We believe this exacerbates catastrophic forgetting and label noise interference. To test this hypothesis, we conducted experiments, the results of which are presented in Figure2.

With the goal of addressing the issues of catastrophic forgetting and label noise in CL-NER (Lecun et al., 1998), we propose SKD-NER. To mitigate the adverse effects of frequent parameter updates due to incoherent optimization in continuous learning, our model first computes an entity classification matrix for each segment of text to be processed, inspired by the multi-headed attention mechanism (Vaswani et al., 2017). This matrix computes a score for the different entity classes to be recognized, thereby converting the entity recognition problem into a binary classification problem. In the CL setting, we enable the model to perform well in multi-label learning on the span classification, while equipping it with knowledge distillation at the span entity level based on the Bernoulli distribution it produces. To address the label noise problem in the incremental learning process, we introduce a reinforcement learning strategy for the student model in the knowledge distillation process (Jiang, 2023), using the most suitable knowledge distillation method for the current student model. For model prediction, we introduce a multi-label classification loss function (Su et al., 2022), which

fits well with our sequence labeling approach. This approach offers several advantages: 1) It allows for knowledge distillation and retention of old knowledge during the CL process. 2) The introduction of reinforcement learning effectively reduces label noise while also addressing the issue of catastrophic forgetting that arises due to frequent weight updates for the same entity in different steps based on different categories. 3) The proposed framework of the model can be used as a reusable framework for continuous learning knowledge distillation and applied to other model migration or continuous learning domains.

We evaluated our model on two Named Entity Recognition (NER) datasets, namely OntoNotes5(Hovy et al., 2006), and Fewnerd(Ding et al., 2021). The experimental results demonstrate that our proposed SKD-NER model significantly outperforms existing continuous learning NER models and achieves a new state-of-the-art (SOTA) performance. Notably, SKD-NER almost eliminates catastrophic forgetting on relatively simple OntoNotes, thereby achieving "continuous learning" in the true sense. Our contributions can be summarized as follows:

- We propose a Continual Named Entity Recognition model and a reinforcement learning-based knowledge distillation framework that ensures the model's effectiveness for continuous learning, which can be further leveraged in other fields of continuous learning or knowledge transfer.

- Innovatively introduce reinforcement learning strategies to support Continual Named Entity Recognition, while optimizing traditional sequence labeling methods and loss functions to address catastrophic forgetting and label noise problems in continuous learning.

- Through extensive experiments, we demonstrate that our approach achieves state-of-the-art performance in Continual Named Entity Recognition and can be seamlessly integrated as a plug-and-play module to further enhance the performance of other Continual Named Entity Recognition models.

## 2 Related Work

### 2.1 Continual Learning NER

Recent research has expanded the application of Continual Learning (CL) from Computer Vision to Natural Language Processing (NLP) tasks, par-

ticularly NER. While most CL-related works in computer vision focus on accuracy-oriented tasks like image classification, their direct application to CL-NER has shown unsatisfactory performance due to challenges in preserving old knowledge in Other-class samples.

Chen and Moschitti (2019) pioneered the study of knowledge transfer in sequence labeling NER models from source to target domains with new entities, using a neural adapter module to handle diverse entity distributions. Following this, AddNER, ExtendNER (Monaikul et al., 2021), and LR (Xia et al., 2022) were developed under a class-incremental setting for CL-NER, employing sequence labeling methods with knowledge distillation. AddNER uses a multi-head approach, while ExtendNER and LR employ single-head layouts with different strategies for handling O tags and old entity mentions. However, current methods such as AddNER and ExtendNER still face forward incompatibility issues and require cumbersome cooperation with knowledge distillation. SpanKL(Zhang and Chen, 2023) explores the potential of span-based models for solving CL-NER with Excellent forward compatibility to solve this problem.

In the context of CL-NER, token-noise is also a very important issue to be concerned about, self-training (Rosenberg et al., 2005; De Lange et al., 2019) has been a straightforward solution for learning old knowledge from Other-class samples. However, this approach suffers from error propagation between models. Recently, Das et al. (2022) have proposed contrastive learning and pretraining techniques to address the problem of token-noise in few-shot NER. CFNER (Zheng et al., 2022) proposes a causal framework in CL-NER that explicitly addresses the challenges posed by token-noise. In contrast, Our approach utilizes reinforcement learning strategies to adjust the process of knowledge distillation, optimizing the process and ensuring the forward compatibility of the model, while also addressing the issue of token-noise to some extent.

## 2.2   Reinforcement learning

Reinforcement learning has found wide application in natural language processing, including machine translation, dialogue systems, and text summarization. More recently, researchers have focused on the potential of RL in the field of continuous learning for natural language processing, with the aim of developing models that can learn new tasks while retaining the memory of previously learned knowledge. For example, Ruder (2019) uses RL to fine-tune pre-trained models with a combination of different reinforcement learning strategies to adapt them to new tasks. Bo et al. (2019) leverages a selector to choose source domain data that is close to the target and accepts rewards from both the discriminator and transfer learning module. Ramamurthy et al. (2022) present NLPO (Natural Language Policy Optimization), a policy optimization-based RL algorithm that dynamically learns task-specific constraints on language distribution.

Inspired by these studies, in this work, we use RL to expertly select appropriate distillation temperature and loss weights for knowledge distillation to support continuous learning. This approach successfully mitigates the impact of catastrophic forgetting on model recognition accuracy during continuous learning.

## 3   Our Approach

In this section, we will first introduce the task setting for Continual Learning Named Entity Recognition (NER). After presenting the overall structure of the SKD-NER model, we provide a detailed explanation of how the model integrates reinforcement learning strategies.

### 3.1   Problem Formulation

Taking into account the non-overlapping nature of entity types in continual learning NER tasks, we follow recent work and propose CL-NER (Xia et al., 2022) in an incremental setting. Given a series of tasks $T_1, T_2, ..., T_n$ and their corresponding training sets $D_1, D_2, ..., D_n$, for each task $T_n$, a new entity type to be recognized and its training set $D_n$ with annotations for the current entity type are defined.

Specifically, we first define a task $T_1$ and train a model $M_1$ on the dataset $D_1$ to recognize the entity type $E_1$. Then, task $T_2$ defines the model $M_2$ to recognize a new entity type $E_2$ on the dataset $D_2$. It is noteworthy that $M_2$ is obtained through knowledge distillation with reinforcement learning based on $M_1$ to achieve the ability to recognize entity type $E_2$ while retaining the ability to recognize entity type $E_1$. Similarly, in the following $n$ incremental steps, we train the previous model $M_n - 1$ on the dataset $D_n$ to obtain the new model $M_n$, while incorporating a reinforcement learning

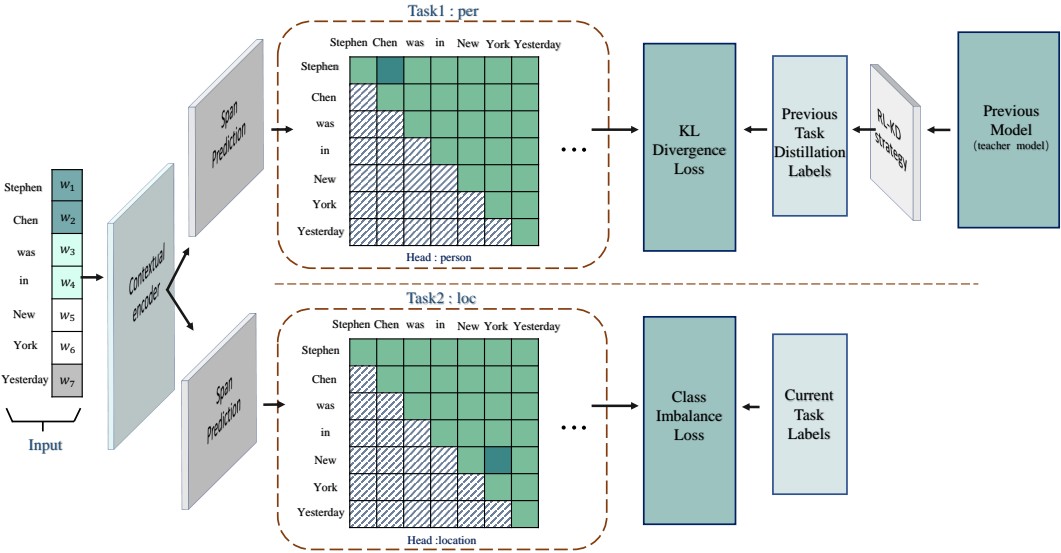

Figure 3: Overall architecture of SKD-NER comprises a shared contextual encoder that is applicable to all tasks and different span representation layers for each entity type in each task. The Bernoulli KL loss and Span loss are calculated on the corresponding entity-relevant span matrices for previously learned entities and currently learned entities, respectively. The RL-KD strategy layer is responsible for identifying the critical parameters for knowledge distillation.

strategy into the knowledge distillation process to ensure the recognition of all entity types defined so far and to reduce catastrophic forgetting and the impact of label noise on the model.

### 3.2 SKD-NER Model

We have introduced an innovative and effective SKD-NER model (refer to Fig. 3) that can learn diverse entity types in a sequential manner for each task. The model takes an input sentence $X$ comprising $n$ tokens: $[x_1, x_2, ..., x_n]$. We define a "span" as a cohesive sequence of tokens that initiate with $x_i$ and culminate with $x_j$, where $1 \leq i \leq j \leq n$. At the $l$-th incremental step, the SKD-NER model endeavors to represent each span in a matrix $h_k$. In this matrix, each span consisting of contiguous tokens $s_{ij}$ is assigned a label corresponding to the current $K$-th entity class. The SKD-NER model consists of the contextual encoder, Span prediction layer, label loss layer, and the RL-KD (reinforcement learning for knowledge distillation) strategy layer.

**Contextual Encoder.** Given an input sentence $X$ comprising $n$ tokens $[x_1, x_2, ..., x_n]$, to capture the dependence between tokens within input sentences, we link each token in $X$ with its corresponding representation in a pre-training language model (e.g., BERT). We define $E = [e_1, e_2, ..., e_n] \in \mathbb{R}^{n \times d_e}$ to represent the embedded vectors of input $X$. After PLM processing, We end up with a new

matrix $H \in \mathbb{R}^{n \times d_h}$, for each token as:

$$h_1, h_2, \ldots, h_n = \text{PLM}(x_1, x_2, \ldots, x_n) \quad (1)$$

**Span Prediction Layer.** Many scholars have thoroughly explored the generation of Span Prediction from tokens and have achieved quite effective results. However, due to the structural bias of Span Prediction which has not been fully understood, Fu et al. (2021) treated span prediction as a system combiner to re-identify named entities from the outputs of different systems. In order to further improve the recognition accuracy of NER models for nested or overlapping discontinuous entities, Zhang et al. (2021) used text syntax dependency to guide the construction of a graph convolutional model to achieve Span Prediction, while Su et al. (2022) employed a multi-head attention mechanism to compute the span matrix. We adopted the latter method, using the starting position token and ending position token of the entity processed by two feedforward layers for dot product calculation to obtain the prediction of the span. Now that we have obtained the representation $h_n$ of the sentence, the process of representing spans can be described as follows:

$$s(i, j) = \textbf{SpanPre}[h_i, h_{i+1}, \ldots, h_j] \quad (2)$$

$$s_a(i, j) = \textbf{FFN}^{i,a}(\mathbf{h}_i)^\top \textbf{FFN}^{j,a}(\mathbf{h}_j) \quad (3)$$

Where $a$ represents the $a$-th type of entity to be identified, $i$ represents the starting position token, and $j$ represents the ending position token. In order to fully utilize boundary information, we introduced the Relative Position Encoding (ROPE) during the span prediction process Su et al. (2022). This encoding explicitly injects relative position information into the model. Specifically, after injecting ROPE position encoding, span $s_a(i,j)$ can be represented as follows:

$$
\begin{aligned}
s_a(i,j) &= \mathbf{FFN}^{i,a}(\mathbf{h}_i\mathbf{R}_i)^\top \mathbf{FFN}^{j,a}(\mathbf{h}_j\mathbf{R}_j) \\
&= \mathbf{FFN}^{i,a}(\mathbf{h}_i)^\top \mathbf{R}_{(j-i)}\mathbf{FFN}^{j,a}(\mathbf{h}_j)
\end{aligned} \quad (4)
$$

**Label Loss Layer.** Building on the span-based method, we generated a marked global upper-triangle matrix for each sentence to be learned. To address this upper-triangle matrix, we devised a scoring function to characterize the relationship between span and the current entity type as follows:

$$
\Omega_{i,j} = \begin{cases} 1 & i \le j \wedge (i,j) \in \mathcal{P}_a \\ 0 & i \le j \wedge (i,j) \notin \mathcal{N}_a \\ -\inf & i > j; \end{cases} \quad (5)
$$

Here, $P_a$ represents the positive set of entities of type $a$ and $N_a$ represents the negative set of entities of type $a$. To enhance the ability of continuous learning NER to recognize entities in the current task, we introduced a span-based cross-entropy loss. This loss not only encourages the model to better learn boundary information but also ensures forward comparability of the model's predictions during the continuous learning process.

$$
\mathcal{L}_{\text{span}} = \log\left(1 + \sum_{1 \le i \le j \le L} \exp\left((-1)^{\Omega_{i,j}} s_a(i,j)\right)\right) \quad (6)
$$

### 3.3 RL-KD strategy layer

To preserve the model's recognition ability for previously learned entity types, we employ knowledge distillation (Gupta et al., 2019; Hinton et al., 2015) to prevent catastrophic forgetting. Specifically, in the $K$-th incremental step $(K > 1)$, we first use the previously learned model $M_k$ (teacher model) to make a one-pass prediction on the entire current training set $D_l$, up to the entity type $E_k$ learned in the previous step of the current task. During this process, we introduce a reinforcement learning strategy to optimize the distillation temperature

(see Fig.4), which will act on the Bernoulli distributions $q_i$ of soft distillation labels for each span of each old entity type. Specifically, during the soft distillation process, the original probability distribution $z_i$ is typically normalized by the softmax and then multiplied by a temperature factor $T$ to obtain a smoothed probability distribution:

$$
q_i = \frac{\exp(z_i/T)}{\sum_j \exp(z_j/T)} \quad (7)
$$

These pseudo-labels are used to compute the Bernoulli KL divergence loss of the current model $M_k + 1$ (student model):

$$
L_{KD}^{\text{SKD}} = KL\left(p_{E^k}^{M_k}, p_{E^k}^{M_{k+1}}\right) \quad (8)
$$

Here, $E^k$ denotes all the learned entity recognition types up to the current step, $p^{M_k}$ represents the soft distillation labels generated by the teacher model, and $p^{M_{k+1}}$ represents the labels produced by the student model for the previous entity types.

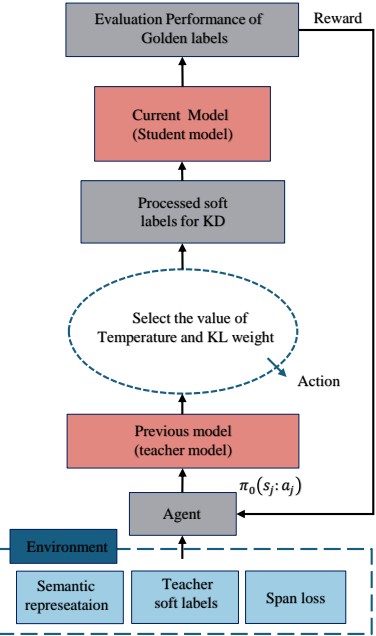

Figure 4: Overall architecture of RL-KD consists of three parts: State, Action, and Reward. This module adjusts the critical parameters of knowledge distillation to optimize model performance.

As with all knowledge distillation works, after one-pass prediction at each step, the final knowledge distillation loss is:

$$
\mathcal{L} = \alpha\mathcal{L}_{span} + \beta\mathcal{L}_{KD} \quad (9)
$$

**State.** Our reinforcement learning approach maintains a sequence of environment states

$s_1, s_2, ..., s_j$. They summarize the input instance and the features of the teacher model, enabling wise decisions to be made accordingly. We design $s_j$ as a vector of real numbers $F(s_j)$, which includes the concatenation of three features. The first feature is the vector representation $R(x_i) \in \mathbb{R}^d$ of the input instance $x_i$. In this paper, we use the score matrix obtained by the span prediction layer as the semantic representation input. The second feature is the prediction by the teacher model $M_k$ on the current input text sequence $x_i$ for all $k$ types already identified before the $(k+1)$-th task. The third feature is the loss of the student model on the input text sequence $x_i$, which is the actual loss of the new entity types that the student model needs to recognize for the current task.

**Action**. The soft labels generated by the teacher model are associated with the distillation temperature and the distillation loss weight during the knowledge distillation process. The agent adjusts the distillation temperature and the weight of the distillation loss for the current teacher model. The policy function $\pi_\theta(s_j, a_j)$ determines a distribution over actions on a state, from which an action value $a_j \in \{0, 1\}$ is sampled. $\theta$ represents the trainable parameters in the policy function.

$$\pi_\theta (s_j, a_j) = [\, \text{temp} \, (s_j, a_j), \, \text{klweight} \, (s_j, a_j)] \quad (10)$$

$$\text{temp} \, (s_j, a_j) = a_j [T + AF(s_j)] + (1 - a_j) T \quad (11)$$

$$\text{klweight} \, (s_j, a_j) = \min [a_j(weight + BF(s_j)) + (1 - a_j) \, weight, 0.1] \quad (12)$$

where $F(s_j) \in \mathbb{R}^{d+(C+1)}$ is the state vector and trainable parameter $\theta = \{A \in \mathbb{R}^{d+(C+1)}, b \in \mathbb{R}^1\}$ The result of the above strategy function definition is an adjusted value assigned to two key parameters in the knowledge distillation process.

**Reward**. The reward function is related to the performance of the student model trained from the distillation of the teacher model. We define a batch of training instances, denoted as $\xi_b = \{x_i, x_{i+1}, \ldots, x_{i+m-1}\}$, where $b$ represents the batch ID and $m$ represents the batch size. For each instance $x_j (i \leq j \leq i + m - 1)$, we construct a state vector $s_{jk}$ for each teacher model $M_k$, and sample an action $a_{jk}$ according to the policy $\pi_\theta(s_j^k, a_j^k)$ (Eq.10). For all sampled $a_{jk}$, we integrate the average of the KL loss (Eq.8) into

the distillation loss KD (Eq.7) to train the student model. To incentivize better model generalization, we use the accuracy metric on the development set $D_0$ and the student model loss as the reward, where $\gamma$ is a hyperparameter balancing the reward from the training set and the development set. Note that the reward is not given immediately after each step is taken. Instead, it is deferred until the completion of the entire batch training.

$$\begin{aligned} reward = & \gamma * (-\mathcal{L}_{CE} - \mathcal{L}_{DL}) \\ & + (1 - \gamma) * \text{Accuracy on } \mathcal{D}_0 \end{aligned} \quad (13)$$

## 4 Experiments

### 4.1 Settings

**Datasets**. We conduct experiments on two widely used datasets, i.e., OntoNotes5 (Hovy et al., 2006), (Ding et al., 2021). Meanwhile, we follow recent works (Monaikul et al., 2021; Xia et al., 2022) to convert the widely used standard NER corpora into separated datasets acting as a series of CL synthetic tasks in class-incremental setting. Both datasets, OntoNotes5 and FewNerd, reflect the model's performance in continual learning tasks to some extent. OntoNotes5 requires only one entity type to be learned for each task, which can reflect the model's performance in simple continual learning tasks. FewNerd, on the other hand, requires multiple entity types to be learned for each task, thereby reflecting the model's performance in more complex continual learning tasks closer to real-world scenarios. We placed the more detailed dataset settings in the appendixA.

**Training**. We use bert-large-cased (Devlin et al., 2018) as the contextual encoder for our model. To split the original training/development set into a series of CL tasks, we randomly divide the samples into unrelated tasks, following previous work (Monaikul et al., 2021). We pre-define a fixed order of classes as the setting for the CL order, as in previous CL work (Hu et al., 2021). However, to avoid excessive randomness in the experiments, we pre-define multiple entity class learning orders for both datasets, and the final experimental results are the average of the results under multiple learning orders. The pre-defined entity class learning orders and more detailed training settings are listed in the appendixB.

**Metrics**. Due to the class imbalance problem in NER, we use Micro F1 and Macro F1 to measure the model performance. We report the Micro F1 and Macro F1 of all learned types up to each step,

| Method | OntoNotes5 | | | | | | |
|---|---|---|---|---|---|---|---|
| | Step1 | Step2 | Step3 | Step4 | Step5 | Step6 | $\delta$ |
| AddNER | 82.53 | 83.65 | 84.46 | 85.02 | 85.43 | 84.97 | 2.44 |
| ExtendNER | 82.69 | 83.16 | 84.34 | 84.54 | 85.06 | 84.83 | 2.14 |
| L&R | 92.06 | 88.09 | 85.69 | 83.79 | 83.38 | 83.02 | -9.04 |
| SpanKL | 85.6 | 87.97 | 88.34 | 88.84 | **88.63** | 88.12 | **2.52** |
| SKD-NER(Ours) | 87.33 | **91.47** | **92.36** | **90.76** | 88.54 | **88.17** | 0.84 |

Table 1: Macro-F1 of different five models at each step on OntoNotes5, and $\delta$ represents the degree of forgetting of the model, which is the difference between the F1 of the final model and the F1 of the first step.

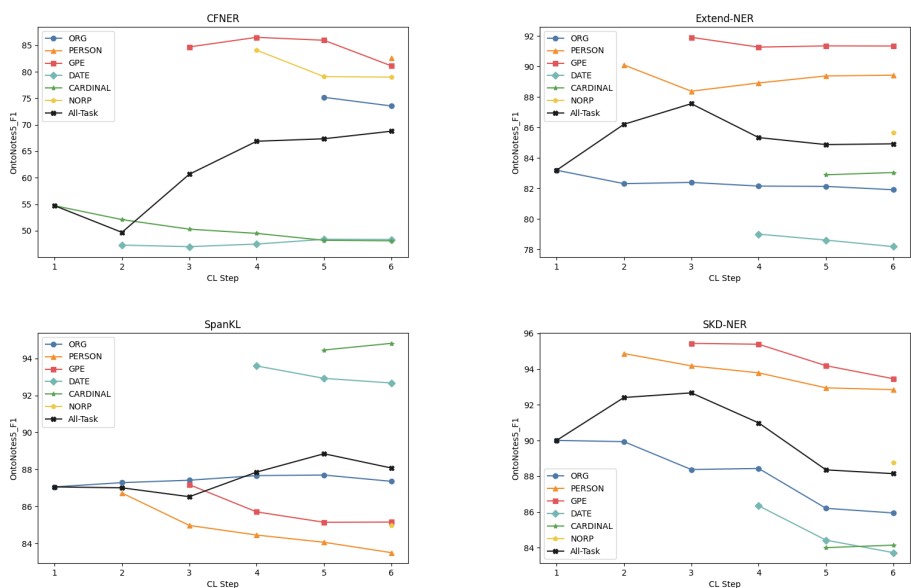

Figure 5: The anti-forgetting performance of four models on various entities in OntoNotes5.

and unless otherwise noted, these results are the average of the results obtained from all pre-defined learning orders.

**Baselines**. We consider five baselines in this work: ExtendNER and AddNER (Monaikul et al., 2021), L&R(Xia et al., 2022), CFNER(Zheng et al., 2022), and SpanKL(Zhang and Chen, 2023). ExtendNER was the previous state-of-the-art method in CL-NER, and L&R was the recent state-of-the-art method in CL-NER. CFNER extracts causal effects in CL-NER tasks and achieves advanced performance in multi-entity type tasks. SpanKL also uses a span-level named entity recognition model for continuous learning, and it performs well in terms of forgetfulness resistan. It is worth noting that these baselines use different dataset splitting methods, and we believe that the dataset splitting method is an extremely important but easily overlooked issue in continual learning tasks. To better approximate real-world scenarios, we randomly divide the samples into unrelated tasks.

### 4.2 Results

**Comparisons with State-Of-The-Art**. We adopt a more realistic dataset scenario by randomly dividing the samples into unrelated tasks, and we compare our method with previous baselines on these datasets. The experimental results on OntoNotes5 are summarized in Table 1, and Figure 5 . In most cases, our method achieves the best performance. In particular, we use a delta value to quantify the difference between the final results of each model's continual learning and the results of the first step of learning, which largely represents the model's anti-forgetting ability. Our model outperforms the previous state-of-the-art by a large margin in this metric. Additionally, in Figure 6, we observe an interesting phenomenon that for a particular entity, the prediction accuracy of our model on distilled models after one or even two steps of distillation is higher than the accuracy on the previous predictions made on the same entity. We attribute this improvement in generalization ability to the knowledge distillation

under reinforcement learning, where future distillation labels may be more accurate and consistent than the originally learned labels. Due to space constraints, we have included the results on the Fewnerd dataset in the appendixD.

**Ablation Study**. We ablate our method, and the results are summarized in Table 2. To validate the effectiveness of the proposed reinforcement learning knowledge distillation method, we also apply this method to traditional sequence labeling methods and BCE-loss methods. Specifically, in w/o SL and w/o SPL, our model still applies the reinforcement learning knowledge distillation method, while in w/o SL RL, we remove both the reinforcement learning policy and the span sequence labeling method. The results show that the reinforcement learning policy plays a significant role in our framework. Additionally, the new sequence labeling method also helps the model further combat catastrophic forgetting. Due to the space limitation of the article, we put the results of FewNerD into the appendix D.

| Method | OntoNotes5 | | | | | |
| --- | --- | --- | --- | --- | --- | --- |
| | Step1 | Step2 | Step3 | Step4 | Step5 | Step6 |
| SKD-NER(Ours) | 87.33 | 91.47 | 92.36 | 90.76 | 88.54 | 88.17 |
| w/o SL & RL | 82.57 | 82.76 | 83.45 | 82.19 | 83.79 | 82.67 |
| w/o SL | 83.69 | 83.59 | 84.17 | 84.26 | 84.76 | 84.43 |
| w/o RL | 86.83 | 86.37 | 86.38 | 85.2 | 85.54 | 85.38 |
| w/o SPL | 86.79 | 91.13 | 92.07 | 90.04 | 88.43 | 88.14 |

Table 2: The ablation study of our method on OntoNotes5, SL: sequence labeling, RL: reinforcement learning strategy, w/o SPL:Replace span loss with BCE loss.

**Anti-CF performance analysis**. We perform Anti-CF performance analysis on the model under OntoNotes5, and the results can be seen in Table 3. For three entities, we track the forgetting situation after 6 steps compared to the initial state. In most cases, our model achieves the best performance. The experimental results demonstrate that our model essentially solves the CF problem on OntoNotes5, which is a relatively simple dataset. We also put FewNerd's results in the appendix D, its results also prove the validity of our method.

**Label noise reduction**. To validate our hypothesis that SKD-NER alleviates the label noise problem in continual learning and leads to improvements, we plot the normalized confusion matrix between different entity types based on the final predictions (Figure 6). Specifically, we use the $'B - X'$ ($X$ denotes a specific entity type) labels in the ground truth as the true labels and the $'B - X'$

| Method | OntoNotes5 | | | | | | |
| --- | --- | --- | --- | --- | --- | --- | --- |
| | | Step1 | Step2 | Step3 | Step4 | Step5 | Step6 | δ |
| ExtendNER | ORG | 83.2 | 82.32 | 82.4 | 82.16 | 82.14 | 81.92 | **-1.28** |
| | PER | 91.31 | 90.1 | 88.38 | 88.92 | 89.38 | 89.43 | -1.88 |
| | GPE | 92.81 | 91.56 | 91.91 | 91.27 | 91.35 | 91.34 | -1.47 |
| SKD-NER (Ours) | ORG | 87.33 | 88.62 | 87.55 | 87.87 | 85.76 | **85.39** | -1.94 |
| | PER | 94.03 | 93.98 | 93.51 | 93.22 | 92.4 | **92.3** | **-1.73** |
| | GPE | 94.64 | 94.79 | 95.16 | 95.2 | 94.06 | **93.4** | **-1.24** |

Table 3: The Anti-CF performance analysis, we analyzed the Anti-CF performance of our model and ExtendNER at each step on three types of entities, and used δ to represent the total forgetting amount.

labels in the model predictions as the predicted labels. From the figure, we can see that compared with ExtendNER, SKD-NER has higher values on the diagonal of the confusion matrix. This indicates that SKD-NER is less affected by incorrectly propagated labels and has more accurate discrimination between different entity types than ExtendNER. These results are consistent with the improvements shown in Table 1.

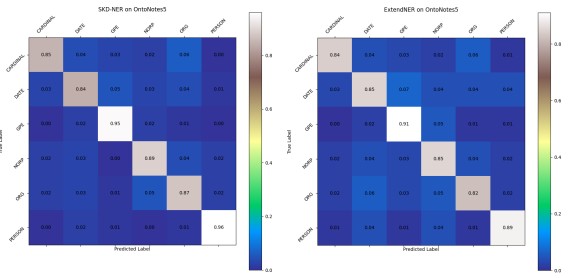

Figure 6: The aconfusion matrix of two models on various entities in OntoNotes5.

## 5 Conclusion

This paper proposes an effective continual learning named entity recognition (NER) model, SKD-NER, aiming to maintain high accuracy for existing entity types while identifying new entity types. We propose a continual learning NER model based on the span method, SKD-NER, which combines reinforcement learning policy. We use knowledge distillation (KD) to preserve memories in the continual learning process and adopt a reinforcement learning policy with a multi-label classification loss for prediction, effectively alleviating the impact of label noise in continual learning. The experimental results demonstrate that the proposed model not only outperforms state-of-the-art methods but also almost solves the catastrophic forgetting problem on OntoNotes5.

# 6 Ethics Statement

For ethical considerations, we provide the following clarifications: (1) We conduct all experiments on existing datasets sourced from public scientific research. (2) We describe the statistical data of the datasets and the hyperparameter settings of our method. Our analysis and experimental results are consistent. (3) Our work does not involve sensitive data or sensitive tasks.

# 7 Limitations

Although the proposed model can partially solve the catastrophic forgetting problem, there is still significant room for improvement in more complex dataset testing. Additionally, due to the introduction of knowledge distillation with a reinforcement learning policy, the training time of our model is slightly longer than that of baselines.

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

# A   Detailed dataset settings

OntoNotes5 annotates 18 entity types, but in practice, to ensure sufficient training samples for each entity type, we filtered out entity types with less than 50 training samples and selected the following types for training: Organization (ORG), Person (PER), Geo-Political Entity (GPE), Date (DATE), Cardinal (CARD), and Nationalities and Religious Political Group (NORP). For FewNerd, we followed recent research to construct each task by using a coarse-grained type, and each task contains multiple fine-grained entity types related to the coarse-grained type. The coarse-grained types include Location (LOC), Person (PER), Organization (ORG), Other (OTH), Product (PROD), Building (BUID), Art (ART), and Event (EVET). Each coarse-grained type contains roughly 10 fine-grained types within it. For example, Product (PROD) contains the following fine-grained entity types: airplane, car, food, game, other, ship, software, train, weapon.

# B   Detailed training settings

The experiments are run on GeForce RTX 3080 Ti GPU. Each experiment is repeated 5 times. During the evaluation process, we only retain the labels of new entity types and set the other labels in the validation set as other classes. For example, we erase the annotations of PER on the samples assigned to the LOC task learning. At each CL step, we select

| **OntoNotes Permutations** |
|---|
| 1 ORG $\rightarrow PER \rightarrow GPE \rightarrow DATE \rightarrow CARD \rightarrow NORP$ |
| 2: DATE $\rightarrow NORP \rightarrow PER \rightarrow CARD \rightarrow ORG \rightarrow GPE$ |
| 3: GPE $\rightarrow CARD \rightarrow ORG \rightarrow NORP \rightarrow DATE \rightarrow PER$ |
| 4: NORP $\rightarrow ORG \rightarrow DATE \rightarrow PER \rightarrow GPE \rightarrow CARD$ |
| 5: CARD $\rightarrow GPE \rightarrow NORP \rightarrow ORG \rightarrow PER \rightarrow DATE$ |
| 6: PER $\rightarrow DATE \rightarrow CARD \rightarrow GPE \rightarrow NORP \rightarrow ORG$ |
| **Few-NERD Permutations** |
| 1: LOC $\rightarrow PER \rightarrow ORG \rightarrow OTH \rightarrow PROD \rightarrow BUID \rightarrow ART \rightarrow EVET$ |
| 2: ORG $\rightarrow PROD \rightarrow ART \rightarrow EVET \rightarrow OTH \rightarrow PER \rightarrow LOC \rightarrow BUID$ |
| 3: PROD $\rightarrow EVET \rightarrow OTH \rightarrow PER \rightarrow ART \rightarrow LOC \rightarrow BUID \rightarrow ORG$ |
| 4: BUID $\rightarrow OTH \rightarrow PROD \rightarrow PER \rightarrow ORG \rightarrow LOC \rightarrow ART \rightarrow EVET$ |

Table 4: Different permutations of tasks used on OntoNotes and Few-NERD.

the model with the best validation performance for testing and the next step of learning. For testing, we retain the labels of all entity types identified until the current task. The pre-defined entity class learning orders are shown in the table 4.

## C Hyper-parameters

In the experiments, we set the dropout rate to 0.1, and we use $d^o = 50$ for all subsequent feed-forward networks in span predictions. For the initial model loss, we set $\alpha = \beta = 1$, and we set the initial distillation temperature to 1. All parameters are fine-tuned using the Adam optimizer (Kingma and Ba, 2015), with a learning rate (lr) of $5 \times 10^{-5}$ for the bert encoder and $1 \times 10^{-3}$ for the remaining networks. After BPE tokenization widely used in PLMs, we limit the maximum sentence length to 256, and we only use the representation of the first subword piece to represent the word after the Bert contextual encoder.

## D Additional Experimental Results

We conducted comparative experiments, ablation studies, and other key experiments on SKD-NER model on the FewNERD dataset. The experimental results demonstrate that our model achieves state-of-the-art performance in mitigating catastrophic forgetting and entity recognition accuracy in CL-NER, outperforming baselines in most cases.

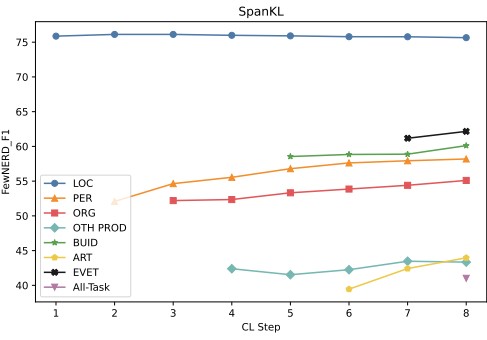

Figure 8: The anti-forgetting performance of SpanKL models on FewNERD.

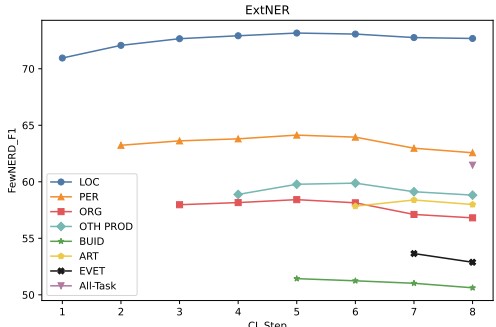

Figure 7: The anti-forgetting performance of Extend-NER models on FewNERD.

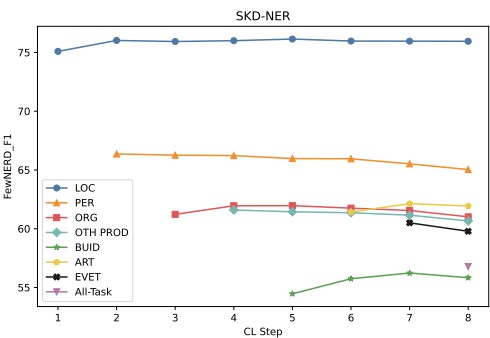

Figure 9: The anti-forgetting performance of SKD-NER models on FewNERD.

| Method | FewNERD | | | | | | | | |
| --- | --- | --- | --- | --- | --- | --- | --- | --- | --- |
| | Step1 | Step2 | Step3 | Step4 | Step5 | Step6 | Step7 | Step8 | $\delta$ |
| AddNER | 64.01 | 62.25 | 61.87 | 61.16 | 61.37 | 62.34 | 63.88 | 63.67 | **-0.34** |
| ExtendNER | 64.06 | 59.02 | 57.03 | 55.79 | 55.65 | 56.03 | 56.78 | 56.09 | -7.97 |
| L&R | 68.13 | 66.72 | 64.51 | 63.44 | 60.97 | 61.23 | 60.88 | 60.32 | -7.81 |
| SpanKL | 67.82 | 64.04 | 63.24 | 62.08 | 64.13 | 61.98 | 63.01 | 62.04 | -5.78 |
| SKD-NER(Ours) | **72.09** | **71.8** | **68.79** | **68.13** | **67.15** | **66.84** | **66.88** | **67.14** | -4.95 |

Table 5: Macro-F1 of different five models at each step on FewNerd, and $\delta$ represents the degree of forgetting of the model, which is the difference between the F1 of the final model and the F1 of the first step.

| Method | FewNERD | | | | | | | |
| --- | --- | --- | --- | --- | --- | --- | --- | --- |
| | Step1 | Step2 | Step3 | Step4 | Step5 | Step6 | Step7 | Step8 |
| SKD_NER(Ours) | 72.09 | 71.8 | 68.79 | 68.13 | 67.15 | 66.84 | 66.88 | 67.14 |
| w/o SL&RL | 64.05 | 59.67 | 58.68 | 57.79 | 56.64 | 56.03 | 56.98 | 57.02 |
| w/o SL | 68.63 | 66.89 | 64.51 | 64.32 | 62.08 | 61.53 | 63.05 | 62.89 |
| w/o RL | 67.76 | 65.23 | 64.43 | 63.08 | 65.19 | 63.27 | 64.66 | 64.07 |
| w/o SPL | 72.08 | 72.01 | 67.63 | 67.56 | 67.02 | 66.69 | 66.74 | 67.03 |

Table 6: The ablation study of our method on FewNERD, SL: sequence labeling, RL: reinforcement learning strategy, w/o SPL:Replace span loss with BCE loss.

| Method | | FewNERD | | | | | | | | |
| --- | --- | --- | --- | --- | --- | --- | --- | --- | --- | --- |
| | | Step1 | Step2 | Step3 | Step4 | Step5 | Step6 | Step7 | Step8 | $\delta$ |
| | LOC | 70.95 | 72.07 | 72.66 | 72.92 | 73.16 | 73.07 | 72.76 | 72.68 | **1.73** |
| ExtendNER | PER | 63.23 | 63.62 | 63.8 | 64.13 | 63.95 | 62.97 | 62.57 | 62.04 | -1.19 |
| | ORG | 57.97 | 58.16 | 58.45 | 58.14 | 57.11 | 56.81 | 56.42 | 55.96 | -2.01 |
| | LOC | 75.09 | 76.02 | 75.93 | 76.0 | 76.14 | 75.97 | 75.96 | **75.95** | 0.86 |
| SKD-NER | PER | 66.36 | 66.26 | 66.22 | 65.97 | 65.95 | 65.52 | 65.03 | **65.18** | **-1.18** |
| | ORG | 61.22 | 61.95 | 61.96 | 61.75 | 61.55 | 61.01 | 61.24 | **61.18** | **-0.04** |

Table 7: The Anti-CF performance analysis, we analyzed the Anti-CF performance of our model and ExtendNER at each step on three types of entities, and used $\delta$ to represent the total forgetting amount.