# OpenReview forum: "SKD-NER: Continual Named Entity Recognition via Span-based Knowledge Distillation with Reinforcement Learning"
_EMNLP/2023/Conference — EMNLP 2023 Main_

### Official Review · Reviewer_8unw · 2023-08-01

**Typos Grammar Style And Presentation Improvements:** 1. Try to avoid situations where ther…
**Soundness:** 3

**Excitement:**

3: Ambivalent: It has merits (e.g., it reports state-of-the-art results, the idea is nice), but there are key weaknesses (e.g., it describes incremental work), and it can significantly benefit from another round of revision. However, I won't object to accepting it if my co-reviewers champion it.

**Missing References:**

Ma, Ruotian, et al. "Learning “O” Helps for Learning More: Handling the Unlabeled Entity Problem for Class-incremental NER." Proceedings of the 61st Annual Meeting of the Association for Computational Linguistics (Volume 1: Long Papers). 2023.

**Paper Topic And Main Contributions:**

This paper introduces a novel Continual Named Entity Recognition (CNER) model and a knowledge distillation framework using reinforcement learning. It combines reinforcement learning strategies with traditional sequence labeling methods to tackle issues like catastrophic forgetting and label noise in continuous learning. Extensive experiments demonstrate that the approach achieves state-of-the-art performance in CNER and can be easily integrated as a plug-and-play module to improve the performance of other CNER models.

**Questions For The Authors:**

The paper only discusses continuous learning for single NER tasks. However, in the era of large language models, a single model can handle multiple tasks with a significant number of parameters. Can the proposed method be applied to continuous learning scenarios with large models? Or does the author have any solutions for continuous learning with large models?

**Reasons To Accept:**

1. The paper is well-written, and the formatting is quite pleasing.
2. The practice of using Reinforcement Learning (RL) to select appropriate distillation temperature and loss weights for supporting continuous learning is quite innovative.
3. The proposed method achieves the state-of-the-art CNER performance and is model-agnostic.

**Reasons To Reject:**

1. The effectiveness of RL-based strategies is often not very stable. And the author did not submit code for review.
2. The experimental results are not rigorous enough as there were no multiple runs for averaging, and no significance tests were conducted.
3. The experimental results comparison in Table 1 is unfair. The ExtendNER [1] and L&R [2] papers use "bert-based-cased" as the encoder, while your SKD-NER paper uses "bert-large-cased" (line 442).
4. The citation format of the paper is not standardized. The formal published version should be cited instead of the Arxiv version. For example, when citing BERT, please ensure to use the appropriate formal publication reference. The authors are requested to double-check all other citations for accuracy.
5. The analysis of the limitations in the paper is too simplistic.


[1] Monaikul, Natawut, et al. "Continual learning for named entity recognition." Proceedings of the AAAI Conference on Artificial Intelligence. Vol. 35. No. 15. 2021.

[2] Xia, Yu, et al. "Learn and review: Enhancing continual named entity recognition via reviewing synthetic samples." Findings of the Association for Computational Linguistics: ACL 2022. 2022.

**Reproducibility:**

3: Could reproduce the results with some difficulty. The settings of parameters are underspecified or subjectively determined; the training/evaluation data are not widely available.

**Reviewer Confidence:**

5: Positive that my evaluation is correct. I read the paper very carefully and I am very familiar with related work.

---

> ### Author Rebuttal · Authors · 2023-08-28
>
> We sincerely appreciate the constructive suggestions from reviewers.
> General response:
>
> Firstly, experiments on the Ontonotes5 and FewNerD datasets demonstrate that our approach achieves state-of-the-art performance in continual learning for named entity recognition on both datasets. Secondly, our main contribution lies in the innovative application of reinforcement learning strategies to the parameter adjustment in the knowledge distillation process. Compared to traditional knowledge distillation methods, our approach has been proven to be more effective in achieving a balance between recognizing new entities and preserving the ability to recognize old entities. Furthermore, we conducted additional experiments to extend our method to other continual learning models.
>
> 1.Code: We will release our code in the near future for experimental purposes and model reproduction.
>
> 2.Experimental results comparison in Table 1: Thank you for pointing out the issue. In Table 1, although ExtendNER and L&R both use bert-base as the encoder, the remaining baselines such as SpanKL use bert-large as the encoder. The experimental results show that our method still achieves state-of-the-art performance. Additionally, we have compared the results of ExtendNER using bert-large as the encoder with our model, and the results are as follows:
>
> |     Method     |       |       | Ontonotes |       |       |       |
> | :------------: | :---: | :---: | :-------: | :---: | :---: | :---: |
> |                | step1 | step2 |   step3   | step4 | step5 | step6 |
> |   ExtendNER    | 82.76 | 83.85 |   84.79   | 84.61 | 85.94 | 85.23 |
> | SKD-NER（ours) | 87.33 | 91.33 |   92.07   | 90.27 | 87.48 | 87.39 |
>
> 3.The citation format of the paper: Thank you for bringing up this matter. We will carefully review and revise the citation format in our paper.
>
> 4.Limitations: In our analysis of limitations, we have identified the presence of forgetting in the model's entity recognition at each step. Although the final model achieves state-of-the-art accuracy, addressing catastrophic forgetting remains a concern that needs to be addressed by the model.
>
> 5.Continuous learning with large models: Our proposed method is an attempt and exploration of applying continual learning techniques to large language models. After conducting thorough experiments, we believe that our approach of using reinforcement learning strategies to adjust key parameters and improve the knowledge distillation process holds significant practical value. We have also conducted related explorations, and knowledge distillation can not only be applied to continual learning with large models but also contribute to model compression, further facilitating the deployment of large models. The previous challenge in knowledge distillation lies in the fixed distillation temperature and distillation loss weight, which lead to severe catastrophic forgetting. In contrast, our proposed method serves as a plug-and-play module that intuitively enhances the performance of the knowledge distillation process and stabilizes the model's memory capacity. We believe this is highly relevant in the era of large language models.
>
> 6.text suggestions: We will revise the teaser texts accordingly.

---

### Official Review · Reviewer_gUVE · 2023-08-03

**Soundness:** 2

**Excitement:**

3: Ambivalent: It has merits (e.g., it reports state-of-the-art results, the idea is nice), but there are key weaknesses (e.g., it describes incremental work), and it can significantly benefit from another round of revision. However, I won't object to accepting it if my co-reviewers champion it.

**Paper Topic And Main Contributions:**

This paper proposes a span-based knowledge distillation approach powered with reinforcement learning to solve continual named entity recognition and catastrophic forgetting issue. Authors argue that their method achieves sota on single and multiple entity types classification on two datasets. Experimental results show that reinforcement learning plays a significant role.

**Questions For The Authors:**

- In section 3.2, hoe many span matrix will be used for the whole continual learning? is there any novelty of span matrix compared to Su et al. 2022?
- In section 3.3, how many RL steps will be used to update the model to reach the expected performance for each entity type classification?
- If the teacher model is not trained well, how do you avoid the training errors transferred to the student model?
- For experimental results, there is only Macro-F1 scores reported in all tables, but the Micro-F1 is not reported according to line 458 and 459. Also the standard division should be better reported, which could better show the effect of multiple ordering and different entity types to the average F1.
- Although different ablation studies are conducted, it is hard to find some deeper analysis or insights from those discussions. Maybe authors could provide some case studies or error analysis of why the proposed method has large performance dropping with steps increasing compared to other baselines.

**Reasons To Accept:**

- a span-based matrix knowledge distillation approach enhanced with reinforcement learning method is introduced
- Achieve sota on two datasets cross single and multiple entity types classification

**Reasons To Reject:**

- there is one contribution mentioned in the introduction, i.e., the proposed method can be applied to other model migration or continual learning domains as a plug-and-play module, which is not proved based on the experimental results.
- Experimental results $\delta$ the degree of forgetting, is not correctly used because the first step only classifies the first entity type, but the final model has already classified all entity types. Then the difference between these two average scores under all entity types and multiple learning orders cannot correctly show whether the model can still forget the earlier entity types. In addition, the value of delta should be the larger the better, to my understanding. However, its value from table 1 and 5 does not show that the proposed approach outperforms other baselines.
- In the figure 5, the proposed model has the largest F1 score performance dropping for almost each entity type compared to other baselines, although each entity type F1 has a higher starting point. This situation seems like that the proposed model does not mitigate the catastrophic forgetting, and it just improves the each entity types F1 at the corresponding step instead. This situation needs a better exploration and explanations.

**Reproducibility:**

3: Could reproduce the results with some difficulty. The settings of parameters are underspecified or subjectively determined; the training/evaluation data are not widely available.

**Reviewer Confidence:**

4: Quite sure. I tried to check the important points carefully. It's unlikely, though conceivable, that I missed something that should affect my ratings.

---

> ### Author Rebuttal · Authors · 2023-08-28
>
> We sincerely appreciate the constructive suggestions from reviewers.
> General response:
> Firstly, experiments on the Ontonotes5 and FewNerD datasets demonstrate that our approach achieves state-of-the-art performance in continual learning for named entity recognition on both datasets. Secondly, our main contribution lies in the innovative application of reinforcement learning strategies to the parameter adjustment in the knowledge distillation process. Compared to traditional knowledge distillation methods, our approach has been proven to be more effective in achieving a balance between recognizing new entities and preserving the ability to recognize old entities. Furthermore, we conducted additional experiments to extend our method to other continual learning models.
>
> 1.In our future work, we expect to further refine our method as a plug-and-play module. This article aims to emphasize the significant contribution of reinforcement learning strategies in improving model accuracy. Currently, we have conducted preliminary experiments to validate the applicability of our method to other continual learning models. Below are the experimental results demonstrating the enhanced performance of the ExtendNER model for continual learning named entity recognition using our method. We have incorporated reinforcement learning techniques to improve its knowledge distillation process while preserving the model's original settings. It can be observed that our method effectively mitigates catastrophic forgetting and enhances the overall named entity recognition accuracy of the final model.
>
> |      Method       |       |       | Ontonotes |       |       |       |
> | :---------------: | :---: | :---: | :-------: | :---: | :---: | :---: |
> |                   | step1 | step2 |   step3   | step4 | step5 | step6 |
> |     ExtendNER     | 82.69 | 83.16 |   84.34   | 84.54 | 85.06 | 84.83 |
> | ExtendNER with RL | 82.69 | 82.09 |   84.56   | 84.61 | 85.35 | 84.94 |
>
> 2.In Table 1, we measure the model's entity recognition capability at each step by taking the average accuracy across all entity types. To validate the degree of forgetting for each entity type, we selected entity types with higher quantities from the dataset (Table 3) for evaluation. The results are as follows. Our method significantly improves catastrophic forgetting in the PER and GPE types. Although our method exhibits a higher degree of forgetting in the ORG type compared to the baseline, our final entity recognition accuracy is significantly higher than the baseline. Similarly, our experiments in Table 1 demonstrate that our method achieves state-of-the-art performance in entity recognition accuracy at the final step of the model.
>
> |  Method   |      |       |       | Ontonotes |       |       |       |       |
> | :-------: | :--: | :---: | :---: | :-------: | :---: | :---: | :---: | :---: |
> |           |      | step1 | step2 |   step3   | step4 | step5 | step6 |   δ   |
> | ExtendNER | ORG  | 83.2  | 82.32 |   82.4    | 82.16 | 82.14 | 81.92 | -1.28 |
> | ExtendNER | PER  | 91.31 | 90.1  |   88.38   | 88.92 | 89.38 | 89.43 | -1.88 |
> | ExtendNER | GPE  | 92.81 | 91.56 |   91.91   | 91.27 | 91.35 | 91.34 | -1.47 |
> |  SKD-NER  | ORG  | 87.33 | 88.62 |   87.55   | 87.87 | 85.76 | 85.39 | -1.94 |
> |  SKD-NER  | PER  | 94.03 | 93.98 |   93.51   | 93.22 | 92.4  | 92.3  | -1.73 |
> |  SKD-NER  | GPE  | 94.64 | 94.79 |   95.16   | 95.2  | 94.06 | 93.4  | -1.24 |
>
> 3.If the delta value is positive, it indicates an improvement in the model's accuracy at the final step compared to the first step. If it is negative, it indicates forgetting compared to the first step. Although our delta values are not the highest in Table 1, our method achieves the highest model performance at the final step, assuming no forgetting occurs in Table 1.
>
> 4.Forgetting in Figure 5: Since our method focuses more on the distillation temperature and weights during the knowledge distillation process, its underlying principle in improving model accuracy lies in balancing the model's ability to learn new entity recognition while preserving the ability to recognize old entities. Due to the influence of temperature adjustment, the model may pay more attention to learning new entities in the next step. However, the final model's recognition accuracy (Table 1) and the degree of forgetting for most individual entities (Table 3) have significantly improved, demonstrating the effectiveness of our method.
>
> 5.Span matrix: For each sentence, we use a span matrix to discriminate each entity type. In comparison to Su et al.'s work, our innovation primarily lies in two aspects: firstly, applying the Span Matrix to the field of continual learning, motivated by the use of the Span Matrix to recognize nested entities and improve model accuracy. Secondly, we introduce reinforcement learning strategies into the knowledge distillation process, thus enhancing the performance of continual learning models.
>
> 6.RL steps: To achieve the desired performance for entity classification, we set 20 batch sizes as one batch, and after each batch training, rewards are given to update the model.
>
> 7.Teacher model: The problem you raised is indeed a major challenge in the knowledge distillation process. There are typically two approaches to address this issue. First, improving the teacher model to enhance its accuracy (Xia et al., 2022). Second, improving the knowledge distillation process. By innovatively employing reinforcement learning strategies to adjust key parameters in the knowledge distillation process, our model can adjust the softness of learning old model labels based on feedback. In Table 1, our initial teacher model accuracy is significantly lower than the L&R model, but ultimately, our model's accuracy is significantly higher than L&R. These experimental results also demonstrate that our proposed method can alleviate this issue to a certain extent.
>
> 8.Micro-F1: Due to the limitations of the article length requirement, we did not report the Micro-F1 scores. This is because in the continual learning problem, we have equal attention to the performance of each category. However, our model also achieves state-of-the-art performance in Micro-F1. Below are some performance results for Micro-F1:
>
> |     Method     |       |       | Ontonotes |       |       |       |
> | :------------: | :---: | :---: | :-------: | :---: | :---: | :---: |
> |                | step1 | step2 |   step3   | step4 | step5 | step6 |
> |   ExtendNER    | 81.39 | 83.04 |   83.87   | 84.45 | 84.98 | 84.54 |
> | SKD-NER（ours) | 87.33 | 91.33 |   92.07   | 90.27 | 87.48 | 87.39 |
>
> 9.text suggestions: Thanks, we will revise the teaser texts accordingly.

---

### Official Review · Reviewer_JuEv · 2023-08-05

**Soundness:** 4

**Excitement:**

3: Ambivalent: It has merits (e.g., it reports state-of-the-art results, the idea is nice), but there are key weaknesses (e.g., it describes incremental work), and it can significantly benefit from another round of revision. However, I won't object to accepting it if my co-reviewers champion it.

**Paper Topic And Main Contributions:**

This paper addresses the issues of catastrophic forgetting and label noise by a reinforcement learning-based knowledge distillation. The reinforcement learning is used to adjust the temperature and the weight of the distillation loss for different entity types.
Experimental results show that the performance is higher than the previous state-of-the-art by 0.05 and 3.47 on two datasets.


**Questions For The Authors:**

Question A: The motivation in the introduction should be better explained. For example, label noise (Line 071) is not explained; Performance comparison (Figure 2) alone is not enough to test the hypothesis.
The necessity of dynamically adjusting the hyper-parameters of distillation is expected.

Question B: Experimental results are not found to support your claim that “our method can be seamlessly integrated as a plug-and-play module to further enhance the performance of other Continual Named Entity Recognition models.”

Question C: Others might have difficulty reproducing the results due to the instability of reinforcement learning. Will you release the code?


**Reasons To Accept:**

This paper introduces a general reinforcement learning-based knowledge distillation to adaptively adjust the hyper-parameters of distillation loss which makes sense.

**Reasons To Reject:**

The motivation should be better organized and explained. The necessity of dynamically adjusting the hyper-parameters of distillation is expected.

**Reproducibility:**

3: Could reproduce the results with some difficulty. The settings of parameters are underspecified or subjectively determined; the training/evaluation data are not widely available.

**Reviewer Confidence:**

4: Quite sure. I tried to check the important points carefully. It's unlikely, though conceivable, that I missed something that should affect my ratings.

**Typos Grammar Style And Presentation Improvements:**

A: The motivation in the introduction should be better explained. See Question A for details.
B: Table 3 is redundant compared with Figure 5. Replacing Table 3 in the analysis of “Anti-CF performance analysis” can better support the claim of “our method almost eliminates catastrophic forgetting on relatively simple OntoNotes”.
C: Upper-bound performance of multitask learning could be added in Table 1 and Table 5 for better comparison.
D: Line 426 missing dataset name
E: Line 483 State-of-the-art
F: Inconsistent nouns e.g. FewNERD, Fewnerd, Few-NERD, FewNerD
G: Title of Figure 6
H: Inconsistent format in Table 4

---

> ### Author Rebuttal · Authors · 2023-08-28
>
> We sincerely appreciate the constructive suggestions from reviewers.
> General response:
>
> Firstly, experiments on the Ontonotes5 and FewNerD datasets demonstrate that our approach achieves state-of-the-art performance in continual learning for named entity recognition on both datasets. Secondly, our main contribution lies in the innovative application of reinforcement learning strategies to the parameter adjustment in the knowledge distillation process. Compared to traditional knowledge distillation methods, our approach has been proven to be more effective in achieving a balance between recognizing new entities and preserving the ability to recognize old entities. Furthermore, we conducted additional experiments to extend our method to other continual learning models.
>
> 1.Motivation
>
> Our motivation for conducting experiments stems from an investigation into the underlying principles of knowledge distillation. The distillation temperature, as a parameter controlling the degree of softening of old entities, and the distillation loss weight, as an indicator balancing distillation loss (retaining the ability to recognize old entities) and model loss (learning new entities), have both been experimentally studied with fixed values in previous work. However, this approach inevitably increases the impact of label noise during the model's knowledge distillation process. Therefore, we made attempts combining reinforcement learning strategies. Experimental results demonstrate that our method effectively improves the model's performance while mitigating catastrophic forgetting.
>
> 2.In our future work, we expect to further refine our method as a plug-and-play module. This article aims to emphasize the significant contribution of reinforcement learning strategies in improving model accuracy. Currently, we have conducted preliminary experiments to validate the applicability of our method to other continual learning models. Below are the experimental results demonstrating the enhanced performance of the ExtendNER model for continual learning named entity recognition using our method. We have incorporated reinforcement learning techniques to improve its knowledge distillation process while preserving the model's original settings. It can be observed that our method effectively mitigates catastrophic forgetting and enhances the overall named entity recognition accuracy of the final model.
>
> |      Method       |       |       | Ontonotes |       |       |       |
> | :---------------: | :---: | :---: | :-------: | :---: | :---: | :---: |
> |                   | step1 | step2 |   step3   | step4 | step5 | step6 |
> |     ExtendNER     | 82.69 | 83.16 |   84.34   | 84.54 | 85.06 | 84.83 |
> | ExtendNER with RL | 82.69 | 82.09 |   84.56   | 84.61 | 85.35 | 84.94 |
>
> 3.Code: We will release our code in the near future for experimental purposes and model reproduction.
>
> 4.Typos Grammar Style And Presentation Improvements: We sincerely appreciate your meticulous review, and we will accordingly examine and amend any grammatical issues in the paper while enhancing the style of expression.

---

### Meta-Review · Area_Chair_hDT3 · 2023-09-14

**Recommendation:** 3

**Metareview:**

The authors consider using reinforcement learning to improve the continual learning for named entity recognition, by using RL to control parameter tuning in a knowledge distillation module for memory preservation, so that new entities can be learned while existing knowledge is also maintained. Results show that the method is effective, and the same method can also be used for other tasks. The content is nicely organized and nicely written.

It would be a nice addition if the authors can discuss further the motivation behind the model design, and give stronger evidence in the stability and significance of the results. The authors promised to release code for RL, which can be useful.

---

### Decision · Program_Chairs · 2023-10-07

**Decision:**

Accept-Main

**Comment:**

The authors consider using reinforcement learning to improve the continual learning for named entity recognition, by using RL to control parameter tuning in a knowledge distillation module for memory preservation, so that new entities can be learned while existing knowledge is also maintained. Results show that the method is effective, and the same method can also be used for other tasks. The content is nicely organized and nicely written.

It would be a nice addition if the authors can discuss further the motivation behind the model design, and give stronger evidence in the stability and significance of the results. The authors promised to release code for RL, which can be useful.